



# Infrared Radiometric Image Classification and Segmentation of Cloud Structure Using Deep-learning Framework for Ground-based Infrared Thermal Camera Observations

Kélian Sommer [1], Wassim Kabalan [2], and Romain Brunet[3]

[1]Laboratoire Univers et Particules de Montpellier, Université de Montpellier, CNRS, Montpellier, France
[2]Université Paris Cité, CNRS, Astroparticule et Cosmologie, F-75013 Paris, France
[3]Aix-Marseille Université, CNRS, CNES, LAM, Marseille, France

**Correspondence:** Kélian Sommer (kelian.sommer@umontpellier.fr), Wassim Kabalan (wassim@apc.in2p3.fr) and Romain Brunet (romain.brunet@lam.fr)

**Abstract.** Infrared thermal cameras offer reliable means of assessing atmospheric conditions by measuring the downward radiance from the sky, facilitating their usage in cloud monitoring endeavors. Precise identification and detection of clouds in images pose great challenges stemming from the indistinct boundaries inherent to cloud formations. Various methodologies for segmentation have been previously suggested. Most of them rely on color as the distinguishing criterion for cloud identification in the visible spectral domain and thus lack the ability to detect cloud structure on gray-scaled images with satisfying accuracy.

In this work, we propose a new complete deep-learning framework to perform image classification and segmentation with Convolutional Neural Networks. We demonstrate the effectiveness of this technique by conducting a series of tests and validations on self-captured infrared sky images. Our findings reveal that the models can effectively differentiate between image types and accurately capture detailed cloud structure information at the pixel level, even when trained with a single binary ground-truth mask per input sample. The classifier model achieves an excellent accuracy of 99% in image type distinction, while the

segmentation model attains a mean pixel accuracy of 94% on our dataset. We emphasize that our framework exhibits strong viability and can be used for infrared thermal ground-based cloud monitoring operations over extended durations. We expect to take advantage of this framework for astronomical applications by providing cloud cover selection criteria for ground-based photometric observations within the StarDICE experiment.

## 1 Introduction

Accurate and continuous monitoring of cloud properties contributes to a profound understanding of atmospheric processes and their subsequent impacts on various Earth systems (Liou, 1992). It provides essential insights for weather predictions and climate dynamics (Hu et al., 2004; Petzold et al., 2015). Observation methods can be divided into two primary distinct categories: downward satellite-based observations (Roy et al., 2017; Martin, 2008) and upward ground-based observations with

all-sky cameras, lidar, radar, and other instruments (Wilczak et al., 1996). The principal aim of satellite-based observations is to investigate the upper regions of clouds, facilitating the examination and analysis of global atmospheric patterns and climate conditions over expansive geographical areas (Schiffer and Rossow, 1983; Boers et al., 2006; Geer et al., 2017; Várnai



and Marshak, 2018). In contrast, ground-based cloud observation excels in the surveillance of localized regions, furnishing valuable data pertaining to the lower segments of clouds by giving information on cloud altitude, cloud extent, and cloud

typology (Bower et al., 2000; Zhou et al., 2019). A combination of these two measurement techniques enhances our overall comprehension of cloud behavior (Mokhov and Schlesinger, 1994; Schreiner et al., 1993; Yamashita and Yoshimura, 2012; Yoshimura and Yamashita, 2013).

Ground-based observations have been extensively used in recent years and have become a viable means to detect, study and identify cloud formations (Paczyński, 2000; Skidmore et al., 2008; Tzoumanikas et al., 2016; Ugolnikov et al., 2017;

Mommert, 2020; Tzoumanikas et al., 2016; Román et al., 2022). As technological evolution has ushered in a new era of monitoring methodologies (Mandat et al., 2014), the utilization of infrared thermal cameras has emerged as a promising avenue for atmospheric investigations through precise radiometric measurements (Szejwach, 1982; Shaw and Nugent, 2013; Liandrat et al., 2017b; Lopez et al., 2017; Klebe et al., 2014; Nikolenko and Maslov, 2021).

Because of their practical use, high sensitivity, low-cost, operating range and wide field-of-view (FOV) (Rogalski, 2011;

Rogalski and Chrzanowski, 2014; Kimata, 2018), it makes them particularly useful for medicine (Ring and Ammer, 2012), agriculture (Ishimwe et al., 2014), aerial (Wilczak et al., 1996), defense (Gallo et al., 1993; Akula et al., 2011), surveillance (Wong et al., 2009), weather forecast (Sun et al., 2008; Liandrat et al., 2017a), or even astronomical related applications to determine the cloud cover fraction during operations and therefore assess the quality of scientific observations (Sebag et al., 2010; Lewis et al., 2010; Klebe et al., 2012, 2014; Reil et al., 2014). Indeed, uncooled infrared microbolometers array sensors

working in the 8-14 μm spectral band can directly detect the long-wave infrared (LWIR) thermal emission of both clouds and the atmospheric background, excluding the scattered light of the sun or starlight (Houghton and Lee, 1972). These LWIR sensors are able to provide high-contrast images and allow fine radiometric measurements to detect low-emissivity cirrus clouds (Lewis et al., 2010; Shaw and Nugent, 2013).

Across recent years, multiple automatic ground-based observation systems have been developed. For example, the infrared

cloud imager (ICI, see Thurairajah and Shaw 2005), can detect clouds and assess cloud coverage both in daylight and at nighttime with a dedicated infrared sensor. Sharma et al. (2015) designed an instrument to detect the cloud infrared radiations to be used in search for a potential site for India's National Large Optical Telescope project. The development of the Radiometric All-Sky Infrared Camera (RASICAM, referenced in Lewis et al. 2010 and Reil et al. 2014) was aimed at enabling automated, real-time quantitative evaluation of nighttime sky conditions for the Dark Energy Survey (Dark Energy Survey Collaboration

et al., 2016). This particular camera is designed to detect, locate, and analyze the motion and properties of thin, high-altitude cirrus clouds and contrails by measuring their brightness temperature against the sky background. The all-sky infrared visible analyzer (ASIVA) is a similar instrument whose primary goal is to provide radiometrically calibrated imagery in the LWIR band to estimate fractional sky cover and sky/cloud brightness temperature, emissivity, and cloud height (Klebe et al., 2014). The ASC-200 system (Wang et al., 2021b) combines information from two all-sky cameras facing the sky operating in both

the visible spectrum (450-650 nm) and the LWIR band.

As next-generation cosmological surveys require more demanding precision on photometric observations – implying better characterization of the atmosphere – monitoring telescope instruments FOV with LWIR thermal cameras may provide sig-



nificant asset to ; (i) classify observations quality in real-time; (ii) evaluate potential cloud coverage (Smith and Toumi, 2008; Liandrat et al., 2017b; Aebi et al., 2018; Wang et al., 2021b); (iii) estimate precipitable water vapor (PWV) content (Kelsey
et al., 2022; Hack et al., 2023; Salamalikis et al., 2023).

In this study, we plan to address the first objective. We use a LWIR thermal infrared camera with a specifically chosen narrower FOV that aims to image the surrounding area of the StarDICE telescope FOV. The StarDICE metrology experiment (Betoule et al., 2022) aims at measuring CALSPEC (Bohlin, 2014) spectrophotometric standard stars absolute flux at the 0.1% relative uncertainty level. Enhanced characterization of atmospheric conditions is required to reach the target sensitivity
(Hazenberg, 2019). As a preliminary step, basic knowledge of the atmosphere conditions in the telescope FOV may provide valuable insights into the quality of spectrophotometric measurements. However, these kinds of infrared instruments operate at high frame rates and produce considerable amounts of data which makes it extremely difficult to analyze by human observers. Therefore, to determine cloud presence in infrared images, deep convolutional neural networks (CNNs) appear to be a viable approach to process images in real-time. Multiple models relying on CNNs have been developed such as CloudSegnet (Dev
et al., 2019a), CloudU-Net (Shi et al., 2021b) CloudU-Netv2 (Shi et al., 2021a), SegCloud (Xie et al., 2020), TransCloud-Seg (Liu et al., 2022), CloudDeepLabV3 (Li et al., 2023), ACLNet (Makwana et al., 2022), DeepCloud (Ye et al., 2017), CloudRaednet (Shi et al., 2022), DMNet (Zhao et al., 2022) and DPNet Zhang et al. (2022). Nonetheless, these methodologies exclusively address RGB-colored images (Li et al., 2011; Dev et al., 2016). Colors or hue provides the essential information for segmentation (especially red and blue channels). In the case of LWIR thermal images, we implement a model capable of
achieving comparable accuracy for single-channel gray-scaled images. Inspired by their large successes in image classification and structure detection for various computer vision tasks, we propose a dedicated deep-learning framework. Our approach is specifically designed towards gray-scaled infrared images and consists of: (i) classifying images (e.g, detecting if any cloud is present in the image and discriminating between clear and cloudy images); (ii) identifying cloud structure (e.g., generating a pixel-based probabilistic segmentation map and verify if the CCD camera FOV is impacted).
The remainder of the paper is structured as follows. Background about the scientific context and related works are presented in Sect. 2. Section 3 details the experimental setup and dataset. Section 4 introduces the proposed framework, describing deep-learning architectures and training procedures. Experimental results and comparisons with other datasets are provided in Sect. 5. Relevant matters and future perspectives are discussed in Sect. 6. Section 7 summarizes the main results and finally concludes the paper.

## 2 Background

### 2.1 Motivation

StarDICE represents one of the initiatives focused on creating a measurement process that bridges the gap between laboratory flux standards (such as silicon photodiodes calibrated by NIST, see Larason and Houston 2008) and the stars found in the CAL-SPEC library of spectrophotometric references (Bohlin et al., 2020). Since type Ia supernovae (SNe Ia) and most astronomical
surveys rely on the calibration of these standard stars for their measurements (Bohlin et al., 2011; Conley et al., 2011; Rubin





et al., 2015; Scolnic et al., 2015; Currie et al., 2020; Brout et al., 2022; Rubin et al., 2022), successfully establishing this connection with high precision effectively addresses the calibration challenge associated with the Hubble diagram for cosmology and the study of dark energy driving the accelerated expansion of the Universe (see Goobar and Leibundgut 2011 for a review of SNe Ia in cosmology).

StarDICE proposal relies on the near-field calibration of a stable light source (Betoule et al., 2022). It serves as a distant in situ reference for a compact astronomical telescope. One of the largest remaining sources of systematic uncertainty when observing stellar sources from the ground is the Earth's atmosphere transmission (Stubbs and Tonry, 2012; Stubbs and Brown, 2015; Li et al., 2016). It is dependent on many environmental conditions and processes, including: absorption and scattering by molecular constituents ($O_2$, $O_3$, and others), absorption by PWV, scattering by aerosols, and shadowing by larger ice crystals
and water droplets in clouds that is independent of wavelength and responsible for gray extinction (Burke et al., 2010, 2017). Current atmospheric transmission or extinction models do not integrate the possible impact of clouds. Indeed, the formation of thin clouds through the condensation of water droplets and ice can result in clouds that are extremely faint and cannot be perceived in the visible spectrum with the naked eye. These clouds often exhibit complex spatial structures, as demonstrated in Burke et al. (2013). Clouds passing through the photometric instrument's FOV result in an attenuation of stellar flux. Previously,
this issue was addressed by incorporating a gray extinction correction, involving the fitting of an empirical normalization parameter for each observation. Nevertheless, as highlighted by Burke et al. (2010), this approach has been proven insufficient due to calibration limitations arising from the dynamic and evolving nature of cloud cover conditions. To tackle this challenge in the StarDICE experiment, our solution involves employing an infrared thermal camera. This specialized equipment offers high-sensitivity radiometric measurements, capturing the sky radiance within the atmosphere's transparency window (10-12
μm). With the help of additional cloud spatial structure identification analytical capabilities, this instrument may be the key to assess photometric observations quality and label science images with superior state-of-the-art accuracy. The primary initial objective is to generate a catalog of optical exposures from the telescope suitable for extracting stellar photometric flux and conducting subsequent analysis.

## 2.2 Related work

In recent years, numerous cloud sky/cloud segmentation algorithms have been introduced along with the increased development of all-sky ground-based cloud monitoring stations (Long et al., 2006; Yang et al., 2012; Krauz et al., 2020; Fa et al., 2019; Mommert, 2020; Li et al., 2022). Indeed, cloud segmentation is a big challenge for remote sensing applications as clouds come in various shapes and forms. Most modern approaches aim to use computer vision algorithms and train them onto very specific publicly available cloud image databases such as: SWIMSEG (Dev et al., 2016), SWINSEG (Dev et al., 2019b, 2017), SWINy-
SEG (Dev et al., 2019a), WSISEG (Xie et al., 2020), HYTA (Li et al., 2011) and TLCDD (TLCDD, 2022). Many proposed solutions are focused on visible RGB images. CloudSegNet (Dev et al., 2019a) is a lightweight deep-learning encoder/decoder network that detects clouds in daytime and nighttime visible color images. CloudU-Net (Shi et al., 2021b) modifies CloudSegNet architecture by adding dilated convolution, skip connection, and fully connected conditional random field (CRF, see McCallum 2012) layers to demonstrate better segmentation performance overall. It uses the powerful U-Net architecture



(Ronneberger et al., 2015) originally applied to medical image segmentation. CloudU-Netv2 (Shi et al., 2021a) replaces the upsampling in CloudU-Net with bilinear upsampling, improves the discrimination ability of features representation, and uses rectified Adam optimizer (rADAM is a variant of the Adam stochastic optimizer (Kingma and Ba, 2014) that introduces a term to rectify the variance of the adaptive learning rate, see Liu et al. 2019). SegCloud (Xie et al., 2020) has been trained onto 400 images and possesses a symmetric encoder-decoder structure and outputs low/high-level cloud feature maps to the same

resolution as input images. TransCloudSeg (Liu et al., 2022) addresses the loss of global information due to the limited receptive field size of the filters in CNN by proposing a hybrid model containing both the CNN and a transformer (Vaswani et al., 2023) as the encoders to obtain different features. CloudDeepLabV3+ (Li et al., 2023) designs a lightweight ground-based cloud image adaptive segmentation method that integrates multi-scale feature aggregation and multi-level attention feature enhancement. ACLNet (Makwana et al., 2022) uses EfficientNet-B0 as the backbone, "à trous spatial pyramid pooling" (ASPP

see Chen et al. 2017) to learn at multiple receptive fields, and global attention module (GAM see Liu et al. 2021) to extract fine-grained details from the image. It provides a lower error rate, higher recall, and higher F1-score than state-of-the-art cloud segmentation models. DeepCloud (Ye et al., 2017) uses the method of Fisher vector encoding which is applied to executing the spatial feature aggregation and high-dimensional feature mapping on the raw deep convolutional features. CloudRaednet (Shi et al., 2022) proposes a residual attention-based encoder-decoder network and trains it over the SWINySEG dataset.

The majority of these models are typically structured using an encoder-decoder architecture, which is the primary innovation brought forth by incorporating CNNs (O'Shea and Nash, 2015). The encoder is tailored to acquire representational features, facilitating the extraction of semantic information while the decoder reconstructs these representational features into the segmentation mask, allowing for pixel-level classification (Badrinarayanan et al., 2017; Alzubaidi et al., 2021).

Others have proposed solutions for all-sky infrared image classification. Liu et al. (2011) applies pre-processing steps
(smoothing noise reduction, enhancement through top-hat transformation and high-pass filtering, and edges detection) before extracting features that are useful for distinguishing cirriform, cumuliform, and waveform clouds. A simple rectangle method as a supervised classifier is applied. They find a 90% agreement between a priori classification carried out manually by visual inspection and their algorithm on 277 images. Sun et al. (2011) suggested: (i) a method for determining clear sky radiance threshold; (ii) cloud identification combined threshold method with texture method; (iii) an algorithm to retrieve cloud base

height from downwelling infrared radiance. They showed that structural features are better than texture features in classifying clouds. Luo et al. (2018) proposed a three-step process: (i) pre-processing; (ii) feature extraction; (iii) classification method to group images into five cloud categories (stratiform, cumuliform, waveform, cirriform and clear) based on manifold and texture features using support vector machine (SVM see Cortes and Vapnik 1995). Their experimental results demonstrate a higher recognition rate with an increase of 2%-10% on ground-based infrared image datasets. These methods classify clouds

into separate categories based on their typology. Until now, all the previously examined approaches, while effective within their specific domains, proved to be unsuccessful when applied to our particular use case. Therefore, we propose a new deep-learning framework based on a linear classifier and U-Net architectures to identify cloud images and detect cloud structures in real-time.



## 3 Experimental setup and datasets

### 3.1 Description of the instrument


Our instrument is an infrared thermal camera similar to the Thurairajah and Shaw (2005) device – specifically the FLIR Tau2 – which operates in the LWIR band, covering the 8-14 μm range. It features a focal plane array (FPA) consisting of 640 × 512 uncooled microbolometers, capturing images at a framerate of 8.33 Hz. The camera is paired with a 60 mm lens at f#1.25 aperture, resulting in a $10.4 \times 8.3$ deg$^2$ imaging area. The primary purpose of deploying this instrument on the equatorial

mount adjacent to the StarDICE photometric telescope is to continuously assess the atmospheric conditions (specifically gray extinction) within the line of sight of the visible CCD camera during observations. The IR instrument FOV is chosen to be larger than the telescope FOV ($0.5 \times 0.5$ deg$^2$) to anticipate the movement of clouds in the smaller FOV of interest. Through meticulous calibration, radiative transfer calculations, and data analysis using simulations, we can extract valuable information about the sky to monitor real-time atmospheric conditions. In Fig. 1, we show the instrument mounted on the equatorial mount

inside the observatory dome, with the necessary command and control equipment. We also monitor the surrounding and internal temperatures of the camera in real-time to correct for temperature-related variations in sensor response. The device is controlled and commanded via the ThermalCapture ThermalGrabber USB 2.0 interface, which grants access to full 14-bit radiometric raw data. We have developed an open-access PYTHON program, available on GitHub[1], to control the camera's functions and capture images. These images are saved in FITS format (Wells et al., 1981). In this study, we only consider raw analog-to-digital

units (ADU) images for simplification purposes but the method would be identical with radiometrically calibrated images.

### 3.2 Datasets and pre-processing

A substantial quantity of images is essential for the effective training and testing of both the classifier and segmentation algorithms. Our dataset comprises LWIR sky images that we captured ourselves. It encompasses a total of 3400 cloudy and clear sky images for the classifier and 4445 sky images with cloudy-only images for the segmentation algorithm and their

associated ground truth masks. To speed up computations and minimize memory consumption, we downsampled the original-sized images into $160 \times 128$ resolution. Cloudy sky images were collected during a three-night period at Observatoire de Haute-Provence (43° 55' 51" N, 5° 42' 48" E) during highly-variable weather conditions. Conversely, cloud-free images were obtained over a shorter time span during the same week.

To compensate for the lack of cloud-free images and prevent potential biases in training due to data imbalance, we generated

synthetic cloud-free images to create a composite dataset containing as many images as the cloudy dataset. These synthetic images replicate realistic observations by simulating 2D horizontal gradients, mimicking the increase in sky downwelling radiance as the camera's field of view tilts toward high zenith angles (i.e. low elevation angles). Realistic sources of noises affecting uncooled infrared thermal cameras are introduced, including: read noise, fixed pattern noise, sky noise, and narcissus effect. This addition ensures that the spatial noise in the synthetic images closely resembles that of actual cloud-free images.

---

[1]https://github.com/Kelian98/tau2_thermalcapture



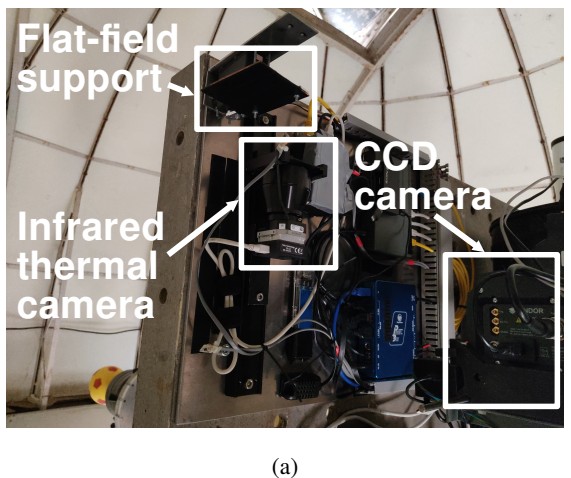

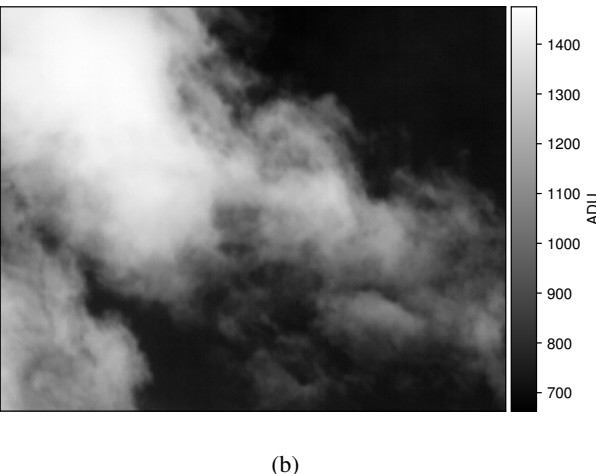

(a)                                                        (b)

**Figure 1. (a)** Infrared instrument installed onto the equatorial table of the StarDICE experiment at Observatoire de Haute-Provence (France), next to the CCD camera and telescope performing photometric measurements of stars. **(b)** Original gray-scale raw infrared thermal image in ADU ($640 \times 512$ pixels).

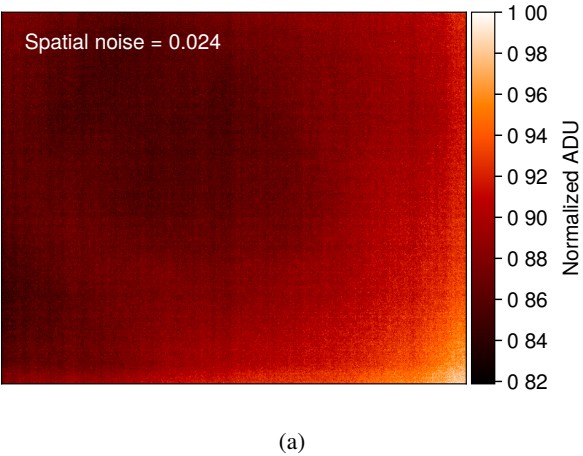

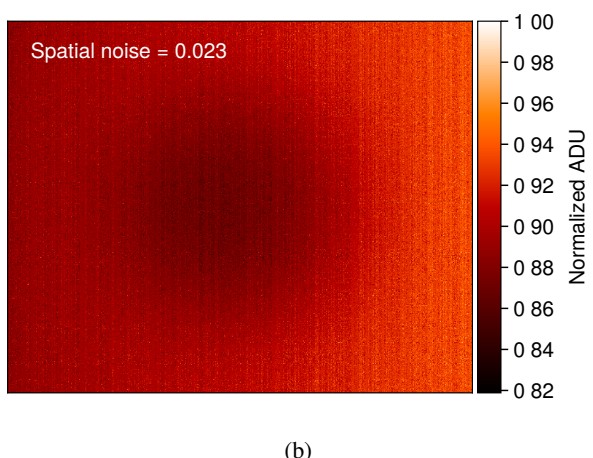

(a)                                                        (b)

**Figure 2.** Comparison of a real observed clear sky image (a) and a synthetically generated realistic image (b). Spatial noise is indicated in the top right corner of each image. Synthetic images demonstrate high fidelity concerning overall spatial noise.

Figure 2 illustrates a typical cloud-free image alongside a synthetically generated one, with spatial noise indicated for each. It's worth noting that the absolute ADU value has no impact, as the data is normalized before training.

All images and masks are visually inspected. Samples presenting artifacts such as tree branches from surroundings or buildings in the FOV corners are discarded. As the camera acquisition framerate enables to get $\sim 8$ images per second, the





pre-processing algorithm included constraints on consecutive image selection based on their time series. Selected frames are
taken from at least 2 seconds between each other to introduce a wider range of displayed clouds.

Ground-truth masks identifying cloud structure on cloud images were manually created through multiple distinct steps of
non-linear stretching procedures using ASTROPY (Astropy Collaboration, 2013, 2018) methods for each image in the dataset.
They consist of a boolean 2D array of the same image size, where *True* identified pixels represent cloud pixels and *False*
identified pixels represent clear sky areas. This step has been partially automated. Binary masks that do not capture the cloud
structure sufficiently have been kept aside to test segmentation model performance. Figure 4 depicts three raw images with
their associated manually generated ground-truth cloud masks for training purposes.

Furthermore, we performed multiple random augmentations (e.g, flip, shear, rotate, shift, and zoom) on each original image
to artificially enlarge the size of each dataset and reduce overfitting (Perez and Wang, 2017; Mikołajczyk and Grochowski,
2018; Yang et al., 2022). All augmented images are produced through the random sequential applications of these five distinct
operations to initial images. These operations are executed with a random varying degree of intensity contained in specific
ranges. Random rotations are applied within an amplitude ranging from -45 to +45 degrees. Shear is introduced with a ran-
dom magnitude ranging from -0.2 to +0.2. Shifting operations are applied between 0 and 50 pixels in both width and height
directions to avoid the generation of unrealistic symmetric structures. Zoom operation is applied within the range of 1 to 3.
No other transformation such as histogram equalization or contrast enhancement is applied to prevent any bias or alteration
in the segmentation performance. After the selection and augmentation procedures, we conducted a visual examination of all
the created sky/cloud images to ensure that they appeared realistic. Since all the parameters in the image augmentation process
undergo controlled adjustments, our generated images closely mirror authentic sky/cloud scenes. Datasets for both models are
split into training and validation subsets with ratios of 80% and 20% respectively.

## 4 Methodology

### 4.1 Overall framework

In this section, we outline the architectural designs of two distinct deep-learning models tailored to automate classification
and segmentation tasks. On the one hand, we implement a linear classifier for image classification, whose specific goal is
to discriminate between cloud-free (photometric) and cloudy images (non-photometric). On the other hand, the segmentation
for cloud structure detection is performed via an optimized U-Net model (Ronneberger et al., 2015) on pre-classified cloudy
images. The output probability map can later be thresholded according to the user's needs to produce the desired predicted
binary pixel segmentation map and allow to obtain finer details regarding the photometric state of the field, at the pixel level.
Figure 3 illustrates the proposed deep-learning framework compared to conventional segmentation algorithms.



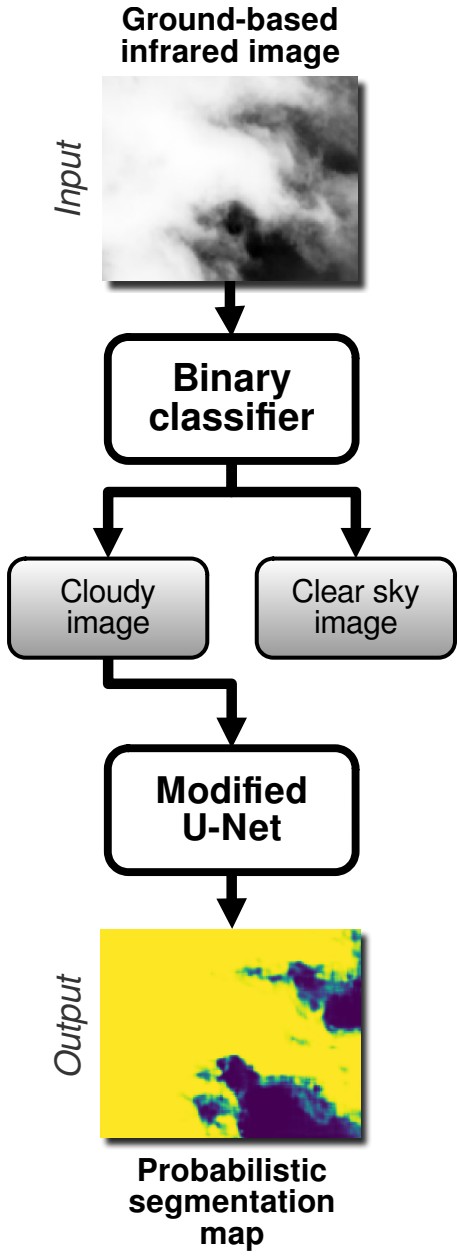

**Figure 3.** Schematic diagram of the framework proposed in this work. An original infrared image goes through the classifier and is labeled as cloudy or clear. Then, the modified U-Net segmentation model identifies cloud structure on the image to finally produce a probabilistic segmentation map that is used to produce reliable metric for our application.



## 4.2 Image classification

For our image classification model, we used a RidgeClassifier from SCIKIT-LEARN (Pedregosa et al., 2011) to classify images

as pure sky (clear) or cloudy. The RidgeClassifier is a linear classification model that employs ridge regression (Hoerl and Kennard, 1970), a technique that introduces a regularization term. This regularization helps in addressing multicollinearity, improving the model's stability and robustness, particularly in scenarios with high-dimensional data or collinear predictors. By balancing the trade-off between bias and variance, the RidgeClassifier effectively minimizes overfitting, making it a suitable choice for our image classification task. The training process for the model involves using a dataset consisting of 2720 images,

where half of them are cloud infrared images and the other half are cloud-free infrared images, each paired with appropriate ground truth labels. The datasets are deliberately balanced to avoid any biases in the model that might favor one class over the other.

## 4.3 Image segmentation

For cloud structure identification, we adopted the U-Net architecture due to its proven efficiency in semantic segmentation

tasks (Ronneberger et al., 2015). The U-Net model comprises an encoder and a decoder, which facilitate the capturing of context-rich features and precise delineation of cloud structures. The encoder employs convolutions and max-pooling layers to progressively downsample the input image, thereby capturing high-level features. These features are then decoded using up-convolutions and skip connections, enabling the accurate reconstruction of the segmented cloud structures. Figure 4 illustrates some examples of infrared cloud images alongside their corresponding ground-truth masks and predictions. Figure 5 depicts

the architecture of the segmentation model.

### 4.3.1 Encoder block

The encoder block of the segmentation model consists of four sets of double convolution blocks (hereafter DoubleConv) and four max-pooling layers. A normalized and binned radiometric image of a fixed input size ($160 \times 128$ pixels) is fed into the model. The DoubleConv contains two sequential convolutional layers, each followed by a Rectified Linear Unit (ReLU)

activation function (Agarap, 2018). The initial DoubleConv block applies a set of learnable filters to the input image, extracting low-level features. Subsequent DoubleConv blocks increase the complexity of the learned features by applying convolutions to the feature map generated by the previous layer, creating a hierarchy of increasingly abstract features. Following each set of DoubleConv blocks, a max-pooling layer is applied to downsample the feature map, reducing its spatial dimensions while retaining the most salient information. The architecture follows a pattern of decreasing spatial dimensions while increasing the

feature depth as we move through the encoder, with the specified channels at each level being 128, 64, 32, and 16 respectively.

### 4.3.2 Decoder block

The decoder block also comprises four sets of DoubleConv blocks, mirroring the encoder structure in reverse order (so in our case 16, 32, 64, and 128). In contrast to the encoder's sequence, which involves a max-pooling layer following each



**Figure 4.** Examples of results with the segmentation model and Otsu's method. Each line represents a different image. The segmentation model displays well-defined cloud structure edges and gives better results than the ground-truth masks and Otsu's algorithm.





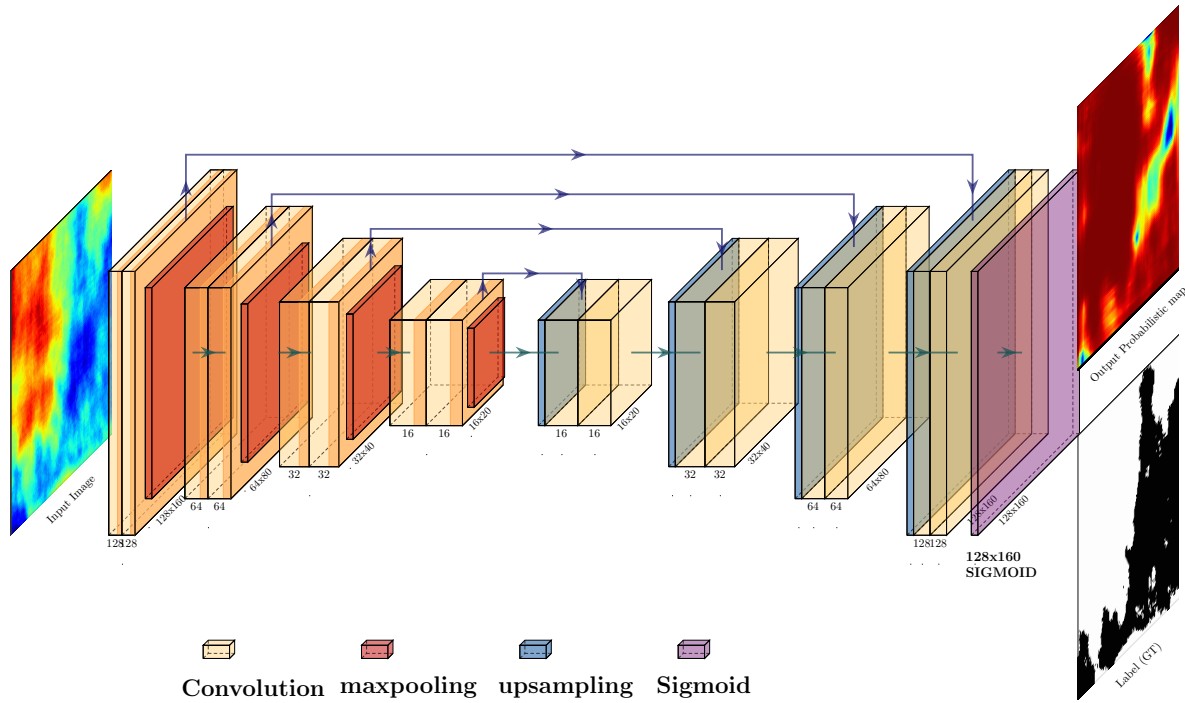

**Figure 5.** Schematic diagram of the proposed U-Net based segmentation model architecture. Yellow boxes represent convolutions. Each double convolution is followed by a Rectified Linear Unit (ReLU) activation function. Each convolution size is indicated on its lower right. The input image is a $160 \times 128$ grayscale image. The output image is a probabilistic mask prediction of pixels being cloudy or clear. Arrows represent operations, specified by the legend-notably, green arrows represent convolutions, while purple ones represent skip connections. Tensor dimensions at the output of each block are specified.

DoubleConv block for downsampling, the decoder employs an upsampling layer preceding each DoubleConv block. The
upsampling operation effectively increases the spatial dimensions of the feature map, preparing it for concatenation with the corresponding, non-downsampled feature map from the encoder provided by the skip connections. Post concatenation, the DoubleConv block is applied to process the merged feature map. This upsampling followed by a convolution is also known as Convolutional Transpose or ConvTranspose. These skip connections ensure coherent and effective feature fusion. This structural configuration is essential for seamlessly integrating both local and global contextual information, thereby improving
the accuracy of segmentation.

### 4.3.3 Model output

The image segmentation model generates a probabilistic mask, assigning a probability value to each pixel, indicating its likelihood of being associated with the cloud category. Using an array of $160 \times 128$ sigmoid functions, the model produces a continuous probability range between 0 and 1 for individual pixels.



### 4.3.4    Fine-tuning the U-Net model

In the original U-Net paper from Ronneberger et al. (2015), a basic convolutional block was interposed between the encoder and
decoder, functioning as a bottleneck to refine feature maps before their upscaling in the decoding path. Yet, through empirical
analysis, we identified that this bottleneck wasn't necessary for our data processing. While many U-Net adaptations utilize a
basic convolutional block for both encoding and decoding, our findings indicated that, during training, the double convolution
blocks outperformed the simple convolution approach.

### 4.4    Training procedure and implementation details

The training process comprises two distinct phases, addressing the classifier model and the U-Net segmentation model. The
loss function employed for training is the binary cross-entropy, which quantifies the difference between predicted probabilities
and actual binary class labels for each instance in the dataset. Mathematically, given an instance's true binary label $y$ (0 or 1)
and the predicted probability $p$ of it belonging to class 1, the binary cross-entropy loss $\mathcal{L}$ is calculated as:

$$\mathcal{L} = -\frac{1}{N}\sum_i y_i \cdot \log\left(f_w(x_i)\right) + (1 - y_i) \cdot \log\left(1 - f_w(x_i)\right) \tag{1}$$

where $\mathcal{L}$ is the binary cross-entropy loss. $N$ is the total number of instances in the dataset, $i$ is the index representing an
individual instance, $y_i$ is the $i$-th true binary label (0 or 1), and $f_w(x_i)$ is the predicted probability that belongs to class 1,
based on the model with parameters $w$. The goal of training is to minimize this loss function by adjusting the model parameters
weights $w$ to better align the predicted probabilities $f_w(x_i)$ with the true labels $y_i$.

Both the classifier and the segmentation algorithms are implemented using the PYTHON programming language, with the
aid of the FLAX package (Heek et al., 2023), a neural network library that is part of the JAX ecosystem (Bradbury et al., 2018).
Training is conducted on the GPU cluster infrastructure of the MESO@LR[2] high-performance computing center, utilizing an
NVIDIA Quadro RTX 6000. To expedite computations and encapsulate the global trend, images are normalized and down-
sampled to the fixed resolution of $160 \times 128$. The models are trained using the ADAM optimizer (Kingma and Ba, 2014)
with a batch size of 64 images. The learning rate is initiated at $\lambda = 10^{-3}$ and decreases with a cosine learning rate decay
function (Loshchilov and Hutter, 2017). To prevent overfitting and expedite the training process, an early stopping mechanism
is employed, which halts the training if the loss value doesn't exhibit a decline below a certain threshold after 15 epochs.

During the training of the U-Net model, the hyperparameter tuning process is carried out to identify the optimal architecture
configuration. This process is facilitated by the OPTUNA framework (Akiba et al., 2019), which employs a sampling strategy
algorithm to explore various configurations. The configurations tested range over different numbers and sizes of filters in the
convolutional layers of the U-Net. Among the numerous configurations tested, the architecture with channels specified as 128,
64, 32, and 16 for the encoder and decoder blocks achieved the lowest loss, indicating superior performance in segmenting
cloud structures.

---

[2]https://meso-lr.umontpellier.fr/





Furthermore, a pruning strategy is integrated within the OPTUNA framework to curtail the exploration of sub-optimal configurations early in the training process, thereby significantly reducing the computational resources and time required for the hyperparameter tuning process. This strategy employs a Median Pruner, which ceases the training of trials exhibiting performance lower than the median performance of completed trials (He et al., 2018; Vadera and Ameen, 2020). The results of the hyperparameter tuning process reveal that the architecture with the specified channels of 128, 64, 32, and 16 outperforms others
in terms of loss minimization.

## 5  Experiments

### 5.1  Performance metrics

In order to evaluate the performance of the proposed models, we adopt several metrics: accuracy (A), precision (P), recall (R), F1-score (F1), the area under the curve (AUC), and the binary cross-entropy loss $\mathcal{L}$ defined in Equation 1. All of these metrics
are defined in the following equations,

$$A = \frac{TP + TN}{TP + TN + FP + FN} \tag{2}$$

$$P = \frac{TP}{TP + FP} \tag{3}$$

$$R = \frac{TP}{TP + FN} \tag{4}$$

$$F1 = \frac{2 \cdot P \cdot R}{P + R} \tag{5}$$

$$AUC = \int_{0}^{1} R(FPR^{-1}(t))dt \tag{6}$$

with True Positives (TP) as the number of correctly classified positive instances; False Positives (FP) as the number of negative instances that were incorrectly classified as positive; False Negatives (FN) as the number of positive instances that were incorrectly classified as negative; True Negatives (TN) the number of correctly classified negative instances; False Positive Rate (FPR) measures the model's ability to incorrectly identify negative instances as positive among all actual negatives and is calculated as FPR = FP / (FP + TN).



## 5.2 Results

### 5.2.1 Classifier

We measure the performance of our models using precision and recall metrics. In this context, precision measures the proportion of correctly predicted cloudy images among all images classified as cloudy. It reflects the classifier's ability to minimize false positives, where a false positive is an instance of predicting an image to contain clouds when it does not. On the other hand, the recall metric quantifies the proportion of actual cloudy images that are correctly identified by the classifier, addressing its capacity to reduce false negatives. A false-negative classification in our case refers to an image that contains clouds but is not recognized as such by the classifier. Table 2 presents the results of the chosen model for the validation subset. All metrics prove the model effectiveness in classifying images, with accuracy, precision, recall, and F1-score all above 99%. The AUC value of 0.99 portrays the classifier as robust. The training process on the entire dataset with ten-fold cross-validation takes under 5 minutes of computing on an average desktop computer.

### 5.2.2 Segmentation

The results presented in Table 1 demonstrate the efficacy of our segmentation model, achieving an accuracy of 94.64% and AUC value of 0.97. Figure 6 depicts the binary cross-entropy loss as a function of training epochs. The shape of the decay in these curves aligns with anticipated training patterns, confirming the model's normal training behaviour. The loss stabilizes around the 300-iteration mark, serving as a benchmark for the model's application.

Figure 7 illustrates the resulting ROC curve. As in Dev et al. (2019a), we adopt a threshold of 0.5, nearly balancing true and false positive rates. However, users can adjust this threshold to meet specific TPR and/or FPR requirements. Figure 4 shows the results for some images of the validation subset. We find excellent segmentation of cloud structure in the infrared images.

## 5.3 Application with cloud counting

In our methodological framework, the initial step involves the application of Otsu's thresholding (Otsu, 1979) to transform infrared sky images into a binary format. This effectively segregates the cloud features from the background, providing a simplified representation where clouds are distinctly highlighted. Following this, the connected component labeling technique, as implemented by the function `skimage.measure.label`, is employed. This function discerns connected regions within the binary array, where connectivity is defined by the presence of adjacent pixels sharing the same value.

This labeling process assigns a unique identifier to each contiguous cloud region, thus enabling an accurate enumeration of individual cloud formations. By comparing this automated count to visual assessments, our analysis revealed a consistent accuracy in the segmented image counts. The segmented images frequently provided counts that closely matched visual estimations, surpassing the performance of raw and binary mask-derived counts, particularly in scenarios where images suffered from high noise levels. Such robustness underscores the advantage of our segmentation approach in providing reliable cloud quantification.




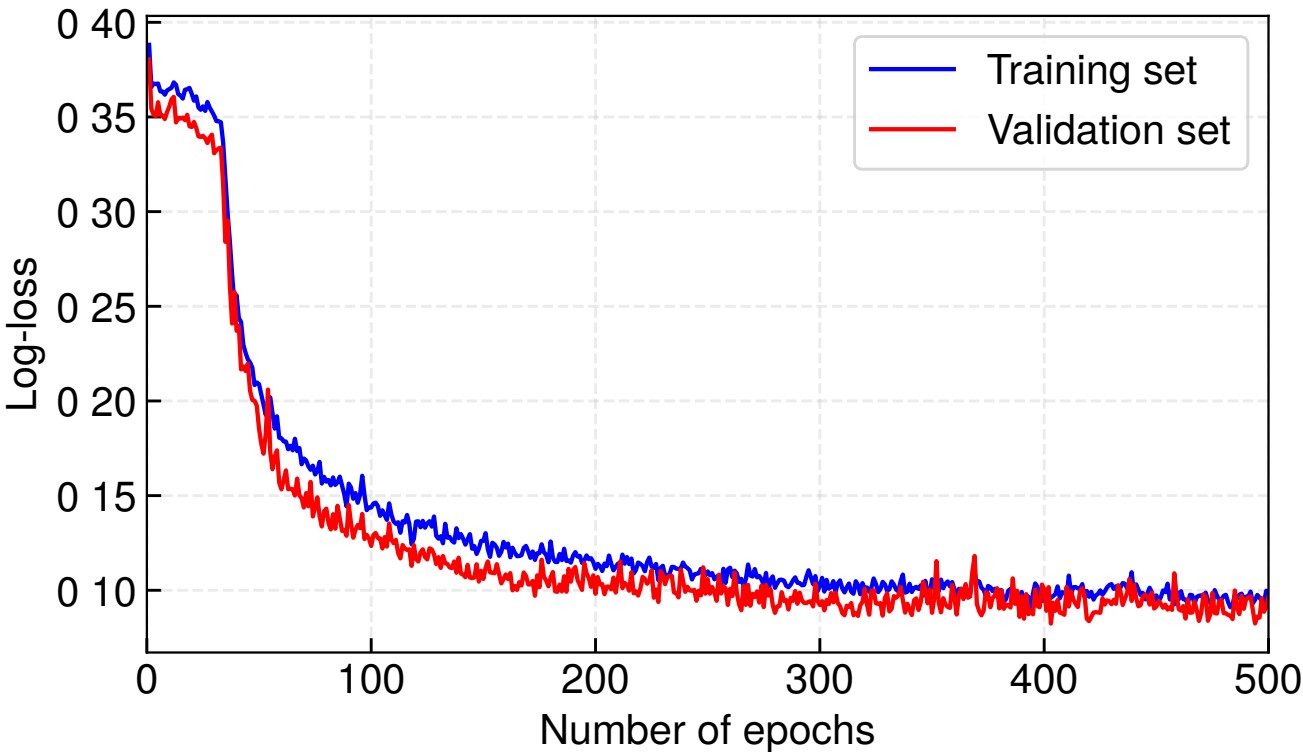

**Figure 6.** Training and testing losses of the segmentation model over epochs. The blue curve is for the training subset whereas the red curve is for the testing subset. After 300 epochs, both curves asymptotically stabilize at a value of approximately 0.10.

## 6 Discussion

### 6.1 Limited benchmarking comparison

Comparing machine-learning models optimized for different types of input data can be meaningful in certain contexts, but it requires careful consideration. The most critical factor is the nature of the data. State-of-the-art models presented in Sect.

2 are optimized for different types of data (3-channel RGB images). It may seem uncoherent to compare the performance with single-channel grayscale infrared images. Indeed, these data types have distinct characteristics, and models may perform differently based on these differences. Some models Sun et al. (2011); Liu et al. (2011); Luo et al. (2018) have been proposed to target infrared images with categorization tasks. Considering whether the models can be adapted or fine-tuned to work with both RGB and infrared data is challenging. This might involve multi-modal learning approaches (Liu et al., 2018; Li et al.,

2020; Wei et al., 2023) or transfer learning techniques (Manzo and Pellino, 2021; Wang et al., 2021a; Zhou et al., 2021) which are not the intended purpose of this work.





**Figure 7.** ROC curve of the segmentation and classification models. The AUC value represents the area under the curve.

Nevertheless, we attempt to evaluate the robustness of our segmentation model by testing its ability to generalize to other datasets including SWIMSEG (Dev et al., 2016), SWINSEG (Dev et al., 2019b, 2017) and WSISEG (Xie et al., 2020)

We transform RGB images into gray-scaled images with OPENCV (Bradski, 2000) color conversion method `COLOR_RGB2GRAY`
defined by the following equation,

$$\text{RGB} \rightarrow \text{Gray} = 0.299 \cdot \text{R} + 0.587 \cdot \text{G} + 0.114 \cdot \text{B} \tag{7}$$

where R, G, and B are respectively red, green, and blue channels of the input color image. Metrics for each dataset are summarized in Table 1.

While the results indicate accurate recognition of most cloud structures in the images by our models, applying a model trained
on our dataset directly to another dataset yields less satisfactory performance due to the suboptimal transformation of color



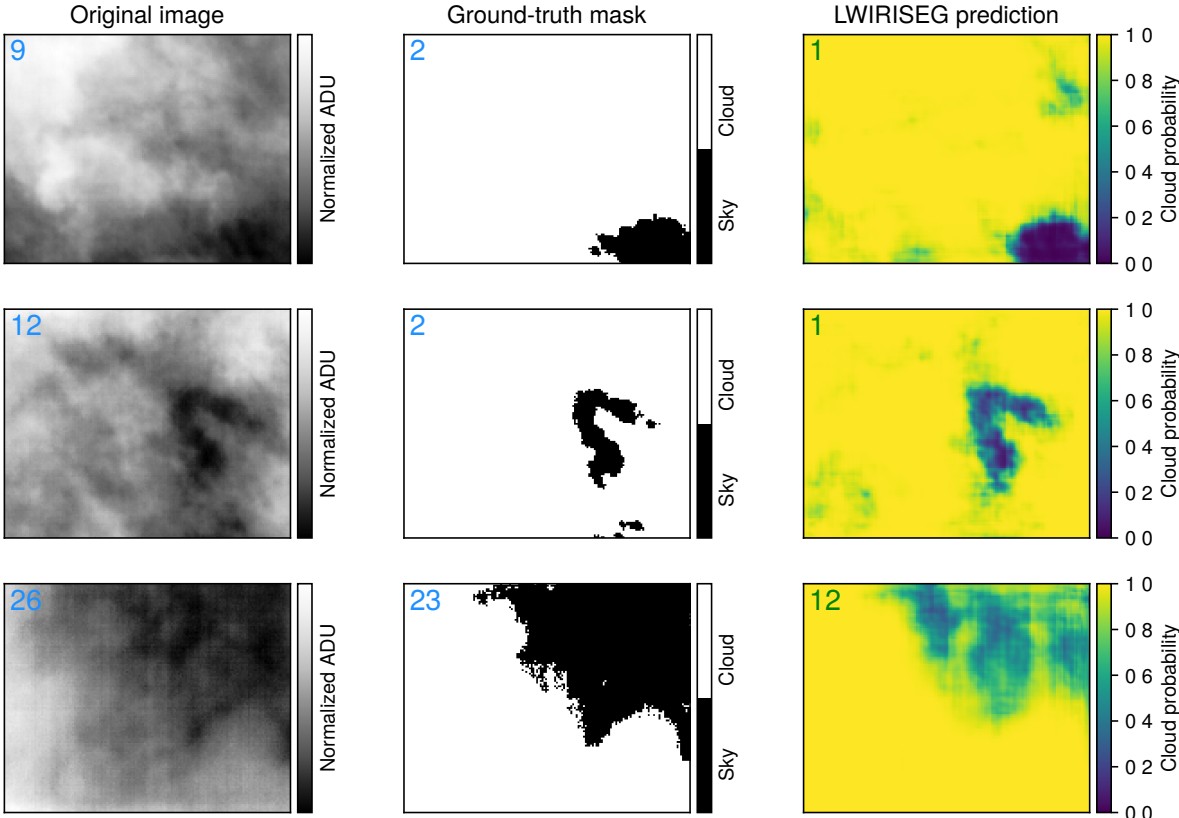

**Figure 8.** Examples of cloud counting different input images. The number of automatically identified clouds is shown in the top-left corner of each subplot. For the upper and center rows, the segmentation map allows the computation of the most accurate result. The last row depicts an example where the computation fails on the original image, the ground-truth mask, and the probabilistic segmentation mask.

images to grayscale. The method's efficiency is hindered by the strong blue color channels resulting from Rayleigh scattering (Bates, 1984), particularly affecting its performance on publicly available datasets transformed in this manner. Further efforts are required to enhance the conversion of RGB color to grayscale images, aiming for comparable contrast to infrared thermal images. Still, our framework demonstrates satisfactory results when exclusively trained on the modified images (FS in Table 1), as opposed to our original grayscale images.


## 6.2 Comparison of segmentation predictions against ground-truth binary mask

The segmentation model occasionally exhibited superior performance compared to the ground-truth masks, as evidenced by examples in Fig. 4. Notably, in this instance, the ground-truth mask failed to identify certain sky patches on the left side of the image. Conversely, the segmentation model's prediction demonstrated a non-zero probability of the existence of sky patches in those areas. This discrepancy arises from the model's utilization of a non-linear mapping technique employing a multi-layer






**Table 1.** Evaluation metrics for the proposed segmentation model on publicly available state-of-the-art datasets. Note that RGB color images are transformed into gray-scale images as the IRIS-CloudDeep segmentation model is optimized for this type of data. Best values are denoted in bold font (A = accuracy, BC Loss = binary cross-entropy loss, AUC = area under the curve). FS describes From Scratch training where as FT means Fine Tuning training.

| Datasets | A [%] | BC Loss | AUC |
|---|---|---|---|
| **LWIRISEG** | **94.64** | **0.1292** | **97.63** |
| SWIMSEG | 57.51 | 6.4259 | 53.32 |
| SWIMSEG FT | 84.56 | 0.3425 | 84.30 |
| SWIMSEG FS | 88.65 | 0.2680 | 95.76 |
| SWINSEG | 67.06 | 4.7272 | 69.55 |
| SWINSEG FT | 91.64 | 0.1999 | 91.29 |
| SWINSEG FS | 93.25 | 0.1670 | 93.64 |

perceptron with ReLU activations. This mapping aims to transform a normalized continuous pixel array into a binary pixel array. Consequently, regions containing sky pixels (denoted by low pixel values) possess the potential to be assigned a low probability value for cloud presence, even when the ground-truth mask assigns certainty (a value of 1) to those patches as cloud-covered areas.

### 6.3 Comparison of segmentation model with Otsu's method

To validate its effectiveness, the segmentation model is evaluated against the conventional Otsu's algorithm with the validation subset from our own LWIRISEG dataset. Otsu's method consists of an adaptative thresholding algorithm that automatically computes the threshold from the image histogram distribution without parameters, supervision or any prior information (Otsu, 1979). Figure 4 depicts some typical comparison results between the two methods and the ground-truth masks given to the deep-learning model for training. The metrics defined in Sec. 5.1 are computed with Otsu's algorithm and presented in Table 2. It demonstrates that the proposed deep-learning modified U-Net model performs significantly better than Otsu's algorithm with mean pixel accuracies being 95.17% and 59.16% respectively. Perfect precision of 100% means Otsu's algorithm does not produce any false positives, implying it is overly conservative in making positive predictions. As noted by Xie et al. (2020), the primary reason for the subpar performance of Otsu's algorithm is its reliance on pixels of the same class having similar gray values, which contradicts the characteristics exhibited by clouds. These experimental findings validate the effectiveness of the segmentation model, highlighting its practical significance for upcoming observations.








**Table 2.** Evaluation metrics for the proposed classification models and comparison between segmentation methods (A = accuracy, P = precision, R = recall, F1 = F1-score, AUC = area under the curve). Best values are denoted in bold font.

| Classification models | A [%] | P [%] | R [%] | F1 [%] | AUC |
|---|---|---|---|---|---|
| **Ridge regression** | **99.26** | **99.31** | **99.28** | **99.28** | **0.99** |
| Logistic regression | 94.85 | 94.95 | 95.25 | 94.85 | 0.94 |
| Perceptron | 93.14 | 93.27 | 93.86 | 93.12 | 0.93 |
| SVM | 91.35 | 91.52 | 92.50 | 91.31 | 0.91 |
| Segmentation methods | | | | | |
| **LWIRISEG** | **95.17** | 96.54 | **97.89** | **97.21** | **0.98** |
| Otsu's algorithm | 59.16 | **100.00** | 52.53 | 68.88 | 0.76 |

## 6.4 Comparison between linear and non-linear methods for classification

In our analysis of infrared sky images, we assessed the suitability of linear classifiers against more complex deep-learning models. Employing linear classifiers such as SVM, Logistic Regression, Perceptron, and Ridge Regression, optimized via
Stochastic Gradient Descent (SGD) with l2 regularization, we aimed to prevent overfitting and maintain model simplicity. Among these, Ridge Regression emerged as the top performer.

Dimensionality reduction techniques supported these findings. Principal Component Analysis (PCA) demonstrated that two principal components could explain a significant portion of the variance (96.1%), implying that the data is almost linearly separable. However, non-linear dimensionality reduction through Uniform Manifold Approximation and Projection (UMAP)
perfectly segregated cloud from cloud-free images, indicating that while the data is nearly linearly separable, non-linear methods provide strong separation as shown in Fig. 9.

In summary, for the task at hand, deep learning models such as ResNet (He et al., 2015) may seem excessive. A well-tuned linear model, particularly Ridge Regression, is equally effective, if not more, due to its interpretability and simplicity. The near-linear nature of the data suggests that simpler models could suffice for such classification challenges.

## 6.5 Future perspectives

The framework established in this paper is one subpart of the StarDICE data processing operations. This will serve the rest of the analysis by identifying and classifying the quality of photometric exposures performed in parallel.

The work undertaken in this paper will be used and the associated module will include the following operations for analysis: (i) classifying infrared sky images obtained by the imaging system in real-time; (ii) analyzing cloud-labeled sky images and



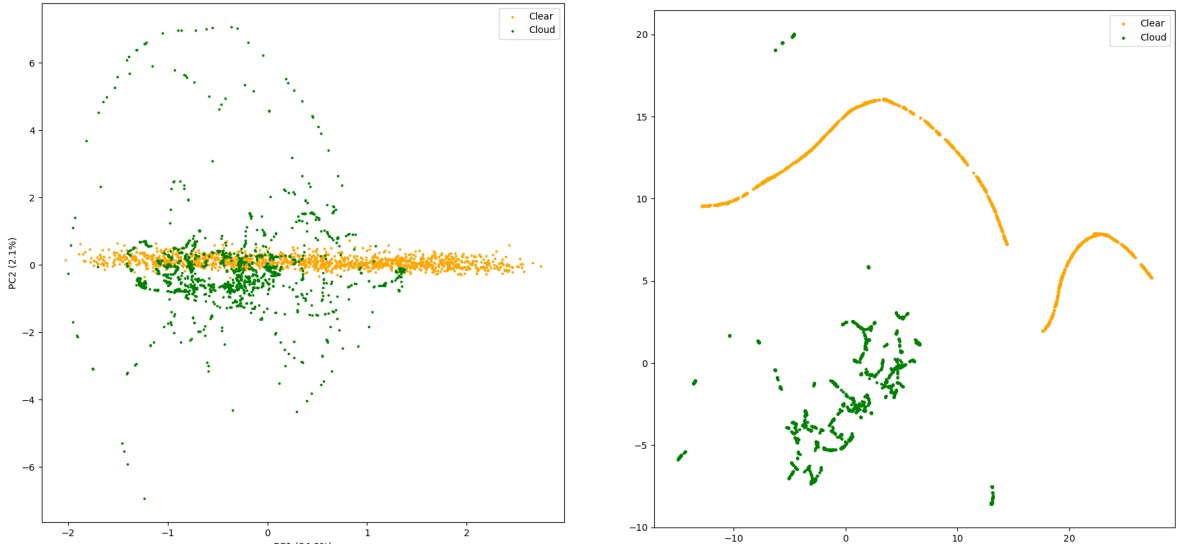

**Figure 9.** Left: first two principal components of the PCA of the entire dataset, representing 96.1% of the variance, with the label clear being the cloud-free images. Right: two-dimension representation of UMAP of the same dataset.

deriving the corresponding cloud structure and cover using the segmentation algorithm; (iii) generating alerts/flag in accordance with the results.

    As mentioned in Sect. 6.1, we were able to train a model on our grayscale images and another model on RGB images, and both models produced great results. However, using a model trained on grayscale images to predict masks for RGB images or even RGB images transformed to grayscale yielded poor results. Additionally, training a model on all the data resulted in
suboptimal performance. As a future endeavor, we can explore the usage of a multimodal deep-learning model that can work with both RGB and grayscale images.

    Improving the accuracy and robustness of our framework could involve further training on a larger dataset in more variable conditions. With the upcoming remote operations capabilities of the telescope system expected to yield a substantial volume of data next year, we anticipate capturing a broader spectrum of sky atmospheric conditions. In this case, a single network could
predict two types of outputs (e.g, pixel segmentation map and a metric describing image quality). Still, additional effort will be necessary to categorize the images based on the varying cloud coverage types.

    Finally, the standard U-Net model lacks temporal correlation. No information about the displacment of the cloud is taken into account. We could gain in model accuracy by incorporating the temporal information using RNN-based algorithms (Sherstinsky, 2020) or temporal U-Net to effectively model temporal information in sequences (Funke et al., 2023).



## 7 Conclusion


In this paper, we proposed a deep-learning framework for the classification and segmentation of ground-based infrared thermal images. As far as we know, it is the first framework that attempts to apply two sequential models for complementary tasks on single-channel gray-scaled infrared images. Specifically, we presented the linear classifier and the U-Net based segmentation model tailored to extract cloud structures on pre-identified cloud images whether during the day or at night. The segmentation

model provides the capability to identify clear sky portions in infrared images, creating a catalog of optical images suitable for photometric measurements and analysis. Extensive experimental results on a combination of self-acquired data and transformed publicly available datasets have demonstrated the effectiveness and performance of the proposed framework. We successfully increased the size of training, testing, and validation subsets with random application of augmentation methods. We developed an accurate simulation tool to produce realistic clear sky images. Some limitations are due to the low amount of strictly

different images in various conditions and errors introduced by ground-truth masks incorrectly labeled manually. Nevertheless, we demonstrated that the segmentation model can rectify poor ground-truth masks and sometimes produce better results. In the future, additional data will be collected by the infrared instrument, capturing various weather conditions. The framework may be re-trained on heavier datasets which will probably increase its accuracy. Furthermore, if enough data is collected with many different cloud categories and proven-to-be accurate radiometric calibration, we will be able to expand the segmentation

model to perform cloud typology through multi-label segmentation. The framework established in this work will serve as a basis for the sky quality assessment and further analysis for the StarDICE metrology experiment.

*Code and data availability.* The source code, datasets and other supporting materials will be made available from the corresponding author Kélian Sommer (kelian.sommer@umontpellier.fr) and Wassim Kabalan (wassim@apc.in2p3.fr) upon request.

*Author contributions.* KS conceived the instrument, collected data, pre-processed the dataset, created ground truths masks for segmentation

and realistic synthetic data for classification. WK, RB and KS designed the framework. WK, RB and KS performed the experiments. KS and WK wrote the paper and collected relevant literature. KS, WK and RB revised the manuscript.

*Competing interests.* The authors declare that they have no conflict of interest.

*Acknowledgements.* We thank Alexandre Boucaud, Johann Cohen-Tanugi and Bertrand Plez for their thoughtful comments and constructive suggestions on the revision of the paper. This work has been realized with the support of MESO@LR-Platform at the University of Mont-

pellier. Some of the results in this paper have been derived using ASTROPY (Astropy Collaboration, 2013, 2018), FLAX (Heek et al., 2023), JAX (Bradbury et al., 2018), MATPLOTLIB (Hunter, 2007), NUMPY (Harris et al., 2020), OPENCV-PYTHON (Olli-Pekka and OpenCV, 2016),



PANDAS (Pandas Development Team, 2020), SCIPY (Virtanen et al., 2020) and SCIKIT-LEARN (Pedregosa et al., 2011). English grammar and syntax have been corrected with the help of AI tools including DeepL and OpenAI ChatGPT.



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
