# Peer review of "Infrared Radiometric Image Classification and Segmentation of Cloud Structure Using Deep-learning Framework for Ground-based Infrared Thermal Camera Observations"

_EGUsphere, 2024_

## Author Response (AR1)

Dear Referees,

Thank you for your thorough reviews of our paper. We appreciate your valuable feedback and have carefully considered all your comments and suggestions that improve the clarity of the paper.

Below, we address your questions and comments.

Best regards,

Kélian SOMMER & Wassim KABALAN

**REVIEWER #1**

*The research address the challenge of improving photometric observation quality in the StarDICE experiment with the help of a long-wave infrared camera to capture sky images. A novel deep-learning framework is proposed, which combines a linear classifier and a U-Net model. The classifier helps differentiate between cloudy and clear sky images whereas the U-Net helps to identify cloud structures. The framework was tested on a self-acquired and several public datasets, and demonstrated effective classification and segmentation, generating catalogs of optical images suitable for photometric analysis despite some limitations.*

*The paper presents a novel and effective method. The paper was mostly well-organized and well-written. The authors provided sufficient results to prove the validity of their work. However, there is room for improvement.*

Aside from the authors' LWIR dataset, some benchmark RGB color image datasets, transformed into gray-scale images for evaluation, were used. The authors are suggested to use at least one or two open-source infrared sky image datasets for evaluation if available.

**> K.S : unfortunately, after doing some research, we could not find any other publicly available dataset with IR sky image. By the way, the other goal of the paper (as well as presenting a new method) was to propose the first dataset of this kind. Therefore, the only comparison we have is to transform an RGB dataset to a gray dataset and to perform computations with it.**

While the classification and segmentation models' performances are presented separately, it is also important to provide an evaluation of the overall framework. The classification model can occasionally fail to classify, which will affect the segmentation model's performance as well.

**> K.S : thanks for this comment. We combined the two processes of the framework in a single "algorithm" to classify and then segment input images. A dedicated notebook is available in the source code for this purpose: https://github.com/ASKabalan/infrared-cloud-detection/blob/main/notebooks/Cloud_classification_segmentation.ipynb**

In both Tables 1 and 2, the proposed model's segmentation performance on the LWIRISEG dataset is shown. In Table 1, the performance is compared on the LWIRISEG dataset alongside other open-source RGB datasets that were modified for this experiment. However, it is unclear why the performance on the LWIRISEG dataset differs between these two tables. Making the points clearer within the text and the caption of the tables are suggested.

**> K.S : thanks for the notice. Indeed, there was a glitch as one row corresponded to the last run of the training of the model whereas the other one was taken at a different epoch. We made the correction.**

**REVIEWER #2**

The discussed topic – screening clouds, aiming at for better astronomical measurements – was interesting and the proposed methods seemed applicable. It must be remembered that the essential goal – ground-based detection/classification of clouds with visible or IR frequencies – is a big challenge as such, regardless of technology used. It stems already from the vague definition of cloud(s); we can imagine a row of meteorologists looking at or measuring clouds – and reporting largely deviating classifications. Especially, borders of fuzzy, smooth clouds like cirrostratus are difficult to detect, even to define. Viewing angle, solar effects and noise cause remarkable problems in recognition.

The methodology presented seems valid. The literature review was impressive. Yet another linguistic check of the text is recommended. The authors mention that AI was used in spell checking. A quick check revealed AI-like oddities in the bibliography as well, like in "Olli-Pekka, H. and OpenCV, t.: OpenCV Python packages"    (OpenCV is not a person, and Olli-Pekka is a first name, not last).

**> K.S : thanks for the notice, we corrected it.**

I was also a bit confused of the use of term "ground truth" as typically refers to most reliable data for comparison, yet probably difficult to collect or produce. Here, it seemed more like a result of an alternative, fast method, that could be hence named something else than ground truth.

**> K.S : We added a precision in the text "By \textit{ground truth}, we refer to the empirically and manually generated masks". As you say, it is difficult to produce a real truth image as the definition of cloud in the image is somewhat empirical. However, we manually created masks and visually inspected them.**

*I am somewhat uncertain about my judgement, as I don't actively follow this topic (cloud classification esp. for astronomy) nowadays.*
*Hence, for a further revision, I recommend sending this article to another reviewer.*